# Clustering of SARS-CoV-2 membrane proteins in lipid bilayer membranes

Joseph McTiernan[1], Yuanzhong Zhang[2], Siyu Li[3], Thomas E. Kuhlman[2], Umar Mohideen[2], Michael E. Colvin[4], Roya Zandi[2], Ajay Gopinathan[1]*

1 Department of Physics, University of California, Merced, California, United States of America,
2 Department of Physics and Astronomy, University of California, Riverside, California, United States of America, 3 Department of Physics and Astronomy, California State Polytechnic University, Pomona, California, United States of America, 4 Department of Chemistry and Biochemistry, University of California, Merced, California, United States of America

* agopinathan@ucmerced.edu

## Abstract

The accumulation of viral structural proteins along the endoplasmic reticulum–Golgi intermediate compartment (ERGIC) membrane drives SARS-CoV-2 self-assembly and budding through interactions among proteins, RNA, and the host membrane. The membrane (M) protein, the most abundant structural component, is thought to interact with other proteins and form clusters that induce membrane curvature and initiate virion formation. However, the relative roles of direct and membrane-mediated interactions between M proteins in this clustering process remain unclear. Here, we combine all-atom molecular dynamics (MD) simulations, continuum modeling, and experiments to demonstrate that M–M interactions alone are sufficient to drive clustering in ERGIC-like lipid bilayers, even in the absence of other proteins or RNA. From MD simulations, we quantify the membrane thinning induced by M proteins and the resulting membrane-mediated interaction energy. Integrating these results into a continuum model that describes the evolution of M protein density on a planar membrane, we identify a critical effective interaction energy required for cluster formation at a given protein density. Comparison with atomic force microscopy (AFM) measurements of M protein clusters enables quantitative estimation of the direct and membrane-mediated interaction energies, revealing that direct M–M interactions dominate through an effective oligomerization energy. Together, these findings establish that M protein interactions are sufficient to drive clustering and provide a quantitative framework for understanding the interplay of direct and membrane-mediated forces in coronavirus assembly and budding.

**Data availability statement:** All data and code used in this paper are available through the Zenodo repository at: https://doi.org/10.5281/zenodo.19614750.

**Funding:** This work was supported by University of California Office of the President UC Multicampus Research Programs and Initiatives, grant M21PR3267 (T.E.K, U.M. M.E.C., R.Z, and A.G.); National Science Foundation RAPID, grant 2034794 (U.M., T.E.K. and R.Z.); National Science Foundation, NSF DMR-2131963 (R.Z. and S.L.); National Science Foundation, NSF-CREST: Center for Cellular and Biomolecular Machines at UC Merced, NSF-HRD-1547848 and NSF-HRD-2112675 (A.G.); National Institutes of Health, NIH G-RISE, T32GM141862 (J.M. and A.G.); National Science Foundation, Center for Engineering Mechanobiology, grant CMMI-1548571 (A.G.); and National Science Foundation, Pinnacles Computing Cluster, NSF-ACI-2019144 (J.M.). The funders had no role in study design, data collection and analysis, decision to publish, or preparation of the manuscript.

**Competing interests:** The authors have declared that no competing interests exist.

## Author summary

Coronaviruses such as SARS-CoV-2 must assemble new viral particles within infected cells before they can spread. This process begins when viral structural proteins accumulate on a specific cellular membrane known as the endoplasmic reticulum–Golgi intermediate compartment (ERGIC). Among these proteins, the membrane (M) protein plays a central role in shaping the virus, yet how M proteins organize into clusters that drive assembly has remained poorly understood. In this study, we combined molecular simulations, theoretical modeling, and experiments to investigate how M proteins interact with each other and the surrounding membrane. We found that M proteins can spontaneously form clusters even in the absence of other viral components, and that direct M–M interactions are strong enough to drive this process. By comparing our model predictions with experimental images, we quantified the strength of these interactions and showed that they outweigh the indirect effects mediated by the membrane. Together, our findings provide a quantitative picture of how M proteins self-organize during coronavirus assembly and reveal a key physical mechanism underlying virion formation.

## Introduction

The assembly of viral components into a complete, infectious particle is one of the most critical steps in the viral life cycle. This process determines the number of virions produced, their structural resilience, and their ability to persist and evolve in the host environment [1,2]. For many RNA viruses, assembly is a highly orchestrated event driven by interactions among structural proteins, the viral genome, and the host cell membrane [3–5]. In some viruses, the genome itself plays an active role in the assembly process, and the structure and stability of the resulting protein shell depend sensitively on the size of the encapsulated cargo [6–8]. In coronaviruses, the unusually large single-stranded RNA genome (≈30 kb) imposes additional constraints on packaging, influencing both the organization of the nucleocapsid and the mechanical stability of the virion. Like many RNA viruses, coronaviruses rely on coordinated interactions among structural proteins, the genome, and a host membrane, but uniquely assemble and bud at the membranes of the endoplasmic reticulum–Golgi intermediate compartment (ERGIC), where their structural proteins and genomic RNA come together to form new virions. The four main structural proteins—spike (S), membrane (M), envelope (E), and nucleocapsid (N)—each play distinct but interdependent roles in this process. The S protein is responsible for binding the virion to the host cell, enabling its entry [9–11]. The M protein defines virion shape, provides the scaffold for assembly, and contributes to maintaining the stability of the surrounding membrane [12–14]. The small E protein acts as an ion channel embedded within the membrane and participates in curvature induction [15,16], while the N protein recruits viral RNA and remains inside the virion, bound to the genome [17–19].

A fully formed coronavirus virion consists of S, M, and E proteins arranged along the viral envelope surrounding the RNA–N complex [9,10]. Outside of the S protein [20], the other three structural proteins do not play a direct role in receptor binding or entry, but are essential for efficient and complete virion formation. Depending on the coronavirus, its replication through assembly and budding along the ERGIC requires M protein and either N protein bound to vRNA, E protein, or both [21–24]. However, in every case, M proteins and their interactions with each other [25,26] are required for complete virion formation [9,14]. Furthermore, M protein is the most prominent structural protein in the virion, and is thought to be responsible for guiding the S [27, 14], N/RNA [28–30], and E protein [14,31] throughout the assembly and budding process. After accumulation in the ERGIC, M proteins interact with S and E proteins along the membrane surface, possibly leading to the induction of membrane curvature through protein clusters. Within the surrounding cytoplasm, N protein recruits viral RNA, which in turn binds to these M protein clusters along the surface of the membrane, believed to introduce additional curvature. After sufficient curvature generation, the M/E/S populated ERGIC membrane curls around the virus' genetic material, leading to the budding of the ~100 nm virion into the ERGIC [32–34]. Here, we focus on the initial formation of SARS-CoV-2 M protein clusters vital for viral assembly.

Recent cryo-EM experiments have revealed that the SARS-CoV-2 M protein forms a homodimer that exists in two distinct membrane embedded conformations: a compact or "short" form and an elongated or "long" form [28,35]. Both forms consist of an N-terminal domain oriented such that it slightly protrudes from a formed virion, an embedded transmembrane domain, and an intravirion C-terminal domain. Cryo-ET and cryo-EM studies of full coronavirus virions showed that the short form is localized to thinner regions of the membrane with lower curvature, whereas the long form is found in regions of higher curvature [13]. It remains to be seen if M protein is the driving component of virion-like curvature induction. Furthermore, using an improved method for synthesizing the M protein, we previously confirmed the thinning effect of the short form by combining all-atom molecular dynamics (MD) simulations with atomic force microscopy (AFM) measurements [36]. We also found that M protein clusters can form within ERGIC-like membranes depending on the local protein density. However, only the short form was observed throughout the study, consistent with previous findings suggesting that the long form requires additional structural protein interactions for stability [13,35,36].

Here, we seek a quantitative understanding of how interactions between M proteins—and their surface density—govern the formation and size of self-assembled clusters. A clearer grasp of how these M protein clusters form is essential for elucidating the mechanisms underlying viral assembly and budding. For example, it remains uncertain whether the membrane-thinning behavior of the M protein facilitates membrane scission during budding or instead promotes lateral assembly [37–39]. Additionally, disrupting the M protein has been shown to inhibit virion production [40,41], potentially due to reduced M protein oligomerization [41]. The relative magnitudes of membrane-mediated and direct M–M interactions, and their respective contributions to clustering, are also unknown. Capturing both the influence of a single protein on the local membrane and the collective behavior of hundreds to thousands of proteins requires a multiscale approach. In this work, we therefore combine all-atom MD simulations, continuum modeling, and experimental measurements to address these open questions.

Initially, using an extended 2 $\mu s$ all-atom MD simulation, we measured the membrane-thinning profile in the vicinity of the M protein and computed a corresponding line tension of approximately 0.10 $k_B T$/nm ± 0.04 $k_B T$/nm. The obtained profile is consistent with our earlier 1 $\mu s$ simulations [36] and with observations of the SARS-CoV M protein [13].

Next, we characterized the early stages of M protein assembly within flat supported membranes using atomic force microscopy (AFM). Following the methods described in [36], we generated AFM images of M proteins embedded in 2.25 $\mu$m × 2.25 $\mu$m supported lipid bilayers that mimic the physiological composition of the ERGIC membrane. From these images, we identified the existence of a critical protein density above which cluster formation occurs and analyzed the distribution of inter-cluster distances.

To understand the respective contributions of membrane-thinning-induced line tension and direct M–M interactions in clustering during the onset of assembly, we turned to analytical modeling. We employed a Cahn–Hilliard framework

[42,43] on a flat membrane - a well-established approach for describing two-component phase separation [44]. Using this model, we analytically and numerically determined how M protein clusters form and evolve as a function of protein area coverage and effective interaction energy. We identified the existence of a critical effective interaction energy for each density, where the effective interaction encompasses all nearest-neighbor forces experienced by an individual protein. By directly comparing model predictions with AFM scans and the line tension obtained from our MD-based membrane-thinning profile, we estimated the effective interaction energy of M proteins in the absence of curvature effects as $\epsilon_m \in [7.8\ k_BT, 9.6\ k_BT]$ and the effective oligomerization energy as $\epsilon_{olig} \in [6.9\ k_BT, 8.9\ k_BT]$. These results suggest that membrane thinning does not play a dominant role in M protein cluster formation, but may instead be more relevant at later stages of assembly and budding. Furthermore, the density fraction required for cluster formation on a flat membrane, $\rho \in [0.118, 0.304]$, together with the existence of a critical effective interaction energy, points to strategies for inhibiting cluster formation - as seen in [40,41]. Such inhibition would suppress the initial step of the assembly and budding process, thereby limiting SARS-CoV-2 replication.

## Materials and methods

### All-atom molecular dynamics

The all-atom MD simulation of the short form embedded in a lipid bilayer was performed using the CHARMM36m force field with the MD package GROMACS, version 2022.3 [45,46]. The CHARMM-GUI input generator was used to set up the simulated system with periodic boundary conditions, and supplied the six steps used for equilibration [47–55]. After equilibration, the system was simulated for 2 $\mu s$ with a timestep of 2 femtoseconds in the NPT ensemble. System temperature was maintained at 303.15 K using the Nose-Hoover thermostat [56,57], with the pressure maintained semi-isotropically at 1 bar in the x-y dimensions and separately in the z-dimension using the Parrinello-Rahman barostat [58,59]. The coordinates were saved once every 50 thousand timesteps, or every 0.1 ns, for a total of 20 thousand frames.

The protein was inserted into the membrane with an orientation and depth that matched other studies [28,35]. This insertion can be seen in Fig 1a and 1b. Only residues 9–204 are accounted for in the short form structure (PDB: 7vgs), with the first eight and last 18 residues excluded [28]. The membrane was composed of Chol 15%; DOPC 45%; DOPE 20%; DOPS 7%; POPI 13% in both leaflets, with the solvent consisting of NaCl at a concentration of 0.15 M and TIP3P water. Initially, the simulation box consisted of a 24.7 nm x 24.7 nm membrane in the x-y plane with at least 5 nm of solvent above and below the protruding protein, yielding a total unit cell thickness of 17.1 nm in the z dimension. However, by the end of the simulation, the membrane size changed to 24 nm x 24 nm, with a z-axis box size of 18.1 nm. Atom counts and box size evolution can be seen in S1 Fig. The final trajectory was reoriented frame-by-frame such that the protein was centered in the box for all frames. While each trajectory was fitted to eliminate protein translation, this was not the case for the rotation of the protein. All-atom simulations were visualized using ChimeraX [60], with the python library MDAnalysis used to analyze the processed trajectory [61,62].

Membrane thickness, defined in Fig 1c, is computationally determined by creating an 18 x 18 square grid in the x-y direction with edges defined by the minimum and maximum phospholipid head position in either direction. At each point in time, these heads are binned, and the average height of the lower and upper leaflets are determined. In the case a bin does not have any head atoms, the corresponding upper and lower leaflets are ignored for that position at the given time. From here, the difference between average height for each leaflet is calculated at every point in time beyond 500 ns, and averaged over time to get the figure shown in Fig 1d.

### Thinning induced line tension

Assuming a symmetric deformation, the membrane thickness profile can be considered the combination of two equivalent monolayers. Adapting the model developed in [63] for determining monolayer height in a transition region between higher and lower membrane thickness due to lipid rafts leads to Eq. 1. In this case, the protein is considered a raft with

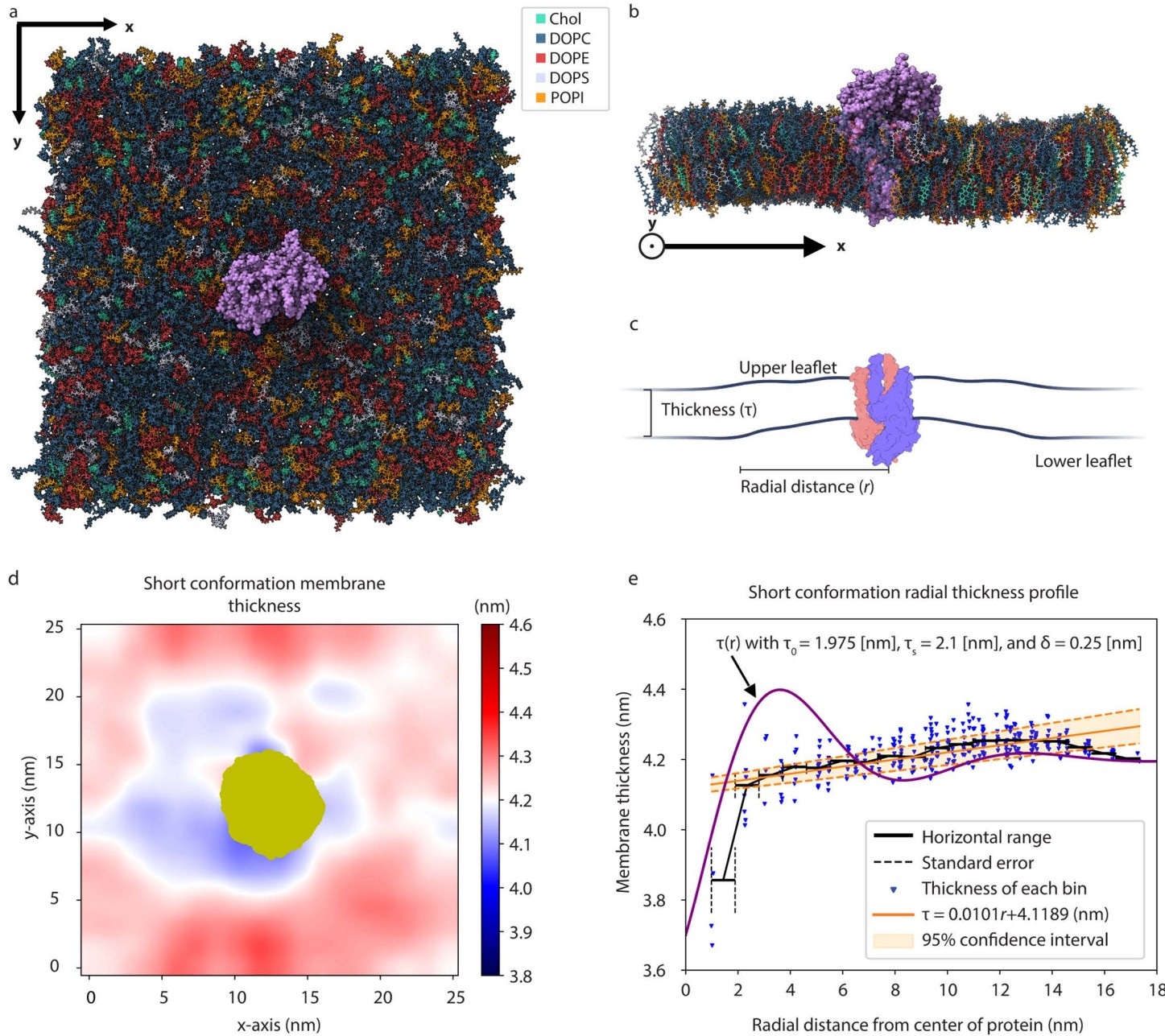

**Fig 1. All-atom molecular dynamics of the M protein short form embedded in a multicomponent membrane.** Final simulation frames of the short form M protein (purple) embedded in a 25 nm x 25 nm lipid bilayer physiologically similar to the ERGIC from above **(a)** and as an x-axis cross-section **(b)**. **(c)** Cartoon defining membrane thickness ($\tau(r)$) as the difference between the upper and lower leaflets for a given radial distance $r$. The two chains in the M protein are distinguished with red or blue, where the C-terminal of the protein pointing downwards is within the virion. Created with BioRender.com (https://biorender.com/se022wy). **(d)** Average membrane thickness is shown along the x-y plane from 500 ns to 2000 ns, where red represents thicker regions of membrane and blue thinner. Gold signifies the cumulative cross-section of the protein. **(e)** Radial thickness profile relative to center of protein, obtained from averaging **(d)** over all angles, is shown with blue representing individual bins. A 95% confidence interval for a line of best fit is shown in orange, with the black trend line representing a radial binning of the blue points. The purple curve represents the analytic solution to the thickness profile given with $\tau_0$ = 1.975 nm, $\tau_s$ = 2.1 nm, and $\delta$ = 0.25 nm.

very high elastic moduli which deforms the surrounding membrane to match its hydrophobic thickness. This expression provides membrane thickness as a function of distance from the protein ($r$), where $\tau_s$ is the thickness of the unperturbed monolayer, $\tau_0$ is the average between low ($\tau_r$) and high thickness monolayers, $\delta$ is the difference between these regions ($\delta = -(\tau_r - \tau_s)$), $\lambda_s = \sqrt{\frac{B_s}{K_s}}$ and $\xi_s = \frac{\sqrt{B_r K_r}}{\sqrt{B_r K_r} + \sqrt{B_s K_s}}\delta$. The factor of two in the equation converts thickness to that of a bilayer, and the contribution from induced curvature is taken to be negligible.

$$\tau(r) = 2\tau_s - 2\xi_s e^{-\frac{\lambda_s}{\tau_0^2}r}\left[\cos\left(\frac{\sqrt{2\tau_0^2 - \lambda_s^2}}{\tau_0^2}r\right) - \frac{\lambda_s}{\sqrt{2\tau_0^2 - \lambda_s^2}}\sin\left(\frac{\sqrt{2\tau_0^2 - \lambda_s^2}}{\tau_0^2}r\right)\right]$$

(1)

In [36] we showed, using AFM methods, that regions with protein are much stiffer than the surrounding membrane. With $B_{s,r}$ and $K_{s,r}$ the bending and tilt moduli of the surrounding membrane and the raft/protein respectively, $B_r \gg B_s$ and $K_r \gg K_s$. Furthermore, Young's modulus measurements of a similar M protein populated supported lipid bilayer system in [36] gives $B_s \sim 3.0\ k_BT \pm 1.0\ k_BT$, leading to $K_s \sim 3.0\ k_BT/\text{nm}^2 \pm 1.0\ k_BT/\text{nm}^2$, $\lambda_s \sim 1$ nm, and $\xi_s \sim \delta$. Error in $B_s$ and $K_s$ are overestimates obtained from standard deviation in Young's modulus measurements from [36].

With this thickness profile, the line tension generated from the cost of bending and tilt can be determined as described in [63]. Without the negligible contributions from induced curvature, Eq. 2 shows the line tension $\gamma_m$. Where $B_r \gg B_s$ and $K_r \gg K_s$ allows for the corresponding approximation, and the factor of two accounts for the bilayer nature of the membrane.

$$\gamma_m = 2\left(\frac{\delta}{\tau_0}\right)^2 \frac{\sqrt{B_s K_s B_r K_r}}{\sqrt{B_r K_r} + \sqrt{B_s K_s}} \approx 2\left(\frac{\delta}{\tau_0}\right)^2 \sqrt{B_s K_s}$$

(2)

## Protein assembly continuum model

To properly represent the assembly of a large number of M proteins along a flat membrane, we utilize a continuum model. We use a Cahn-Hilliard model [42,43] to describe protein density evolution on a flat plane, an approach commonly used to describe two component phase separation [44]. It is to be noted that such approaches have also been extended to describe the motion of curvature inducing transmembrane proteins [64–68], though here we consider only the planar lipid bilayer case and do not consider density dependent membrane surface tension nor membrane viscosity [65,66,69–71]. Our analysis is restricted to a flat membrane for direct comparison with supported lipid bilayer AFM profiles, which show no height variation beyond protrusions from M protein C-terminals. The total free energy of the system consists of both an enthalpic component, arising from protein-protein and protein-membrane interactions, and an entropic part,

$$\mathcal{F} = \mathcal{F}_{entropic} + \mathcal{F}_{interaction}.$$

(3)

The entropic portion of the free energy is a typical entropy of mixing which can be expressed as,

$$\mathcal{F}_{entropic} = \int_{\mathcal{S}} \frac{k_B T}{a^2}\left[(1-\rho)\ln(1-\rho) + \rho\ln(\rho)\right]dxdy,$$

(4)

where $\rho$ is the protein density fraction defined as a two dimensional scalar field along a flat infinitesimally thin surface, $T$ is the temperature, and $a$ represents the nearest neighbor distance between proteins [44]. Additionally, $a$ defines the M protein saturation density ($\rho_s = \frac{1}{a^2}$) [65,68]. We take this distance to be the approximate width of a single M protein since AFM images show tight packing.

The interaction portion of the free energy, $\mathcal{F}_{interaction}$, involves three different terms,

$$\mathcal{F}_{interaction} = \int_{\mathcal{S}} \left[ \frac{\epsilon_m}{2a^2} \rho - \frac{\epsilon_m}{2a^2} \rho^2 + \frac{\epsilon_m}{4} \left| \nabla \rho \right|^2 \right] dxdy \tag{5}$$

where $\epsilon_m$ is an effective interaction energy accounting for all nearest neighbor interactions a single M protein would experience (refer to Effective interaction energy for details) [44]. An effective interaction energy greater than zero implies attraction. This has the form of a two-component regular solution or Flory-Huggins model [42,44,72,73], with volume exclusion, protein attraction, and interfacial energy terms.

Following a standard approach [65,67], the dynamics of $\rho$ can now be expressed using Model B dynamics to account for the conservation of protein density, leading to

$$\frac{\partial \rho}{\partial t} = \frac{Aa^2}{k_B T} \nabla \cdot \left[ \rho \nabla \left( \frac{\delta \mathcal{F}}{\delta \rho} \right) \right]. \tag{6}$$

The full expression for the variational derivative $\frac{\delta \mathcal{F}}{\delta \rho}$ and density evolution can be seen in S1 Appendix.

## Linear stability analysis

To quantitatively understand the conditions and parameter regimes required for clustering, we perform linear stability analysis. This is done by first applying a small perturbation to a homogeneous density state of the form,

$$\rho(x, y, t) = \rho^* + \delta\rho(x, y, t), \tag{7}$$

$$\delta\rho = C_\rho e^{(\omega_0 t - i\mathbf{q} \cdot \mathbf{r})}. \tag{8}$$

Here, the initial protein density fraction is $\rho^*$, and the perturbation $\delta\rho \ll 1$ (set by $C_\rho \ll 1$). Each mode is defined by its wavevector $\mathbf{q}$ and growth rate $\omega_0$. Throughout this work, $\mathbf{q}$ is nondimensionalized according to $\mathbf{q} \to \frac{\mathbf{q}}{a}$. A positive growth rate for any $q$ implies an unstable mode with that wavevector and indicates clustering with an average distance between cluster formations ($d$) set by $q = \frac{2\pi a}{d}$. A schematic characterizing protein assembly in an unstable regime can be seen in Fig 3.

Using Eq. 7 in Eq. 6 and ignoring all nonlinear terms leads to the protein evolution equation,

$$\frac{\partial \rho}{\partial t} = \frac{A\rho^*}{a^2} \left[ \Phi \tilde{\nabla}^2 \rho - \frac{\tilde{\epsilon}_m}{2} \tilde{\nabla}^4 \rho \right], \tag{9}$$

$$\Phi = \frac{1}{\rho^* \left( 1 - \rho^* \right)} - \tilde{\epsilon}_m, \tag{10}$$

where tildes signify nondimensionalized quantities ($\tilde{\epsilon}_m = \frac{\epsilon_m}{k_B T}$ and $\tilde{\nabla} = a\nabla$). Applying the form of the perturbation in Eq. 8 to Eq. 9 allows us to obtain the growth rate as a function of wavevector,

$$\omega_0 = \frac{-A\rho^*}{a^2} q^2 \left[ \Phi + \frac{\tilde{\epsilon}_m}{2} q^2 \right]. \tag{11}$$

From this dispersion relation, the wavevector and growth rate corresponding to the fastest growing modes are determined.

## Effective interaction energy

The effective interaction energy ($\epsilon_m$) (or Flory Huggins interaction parameter) is determined by considering a lattice where lattice sites can be occupied by membrane or a protein, and only nearest neighbor and spatially independent interactions are considered. As such, $\epsilon_m$ can then be expressed in terms of membrane and protein interactions as [44],

$$\epsilon_m = z \left( \epsilon_{m-m} + \epsilon_{mem-mem} - 2\epsilon_{m-mem} \right),$$ (12)

where $z$ defines the number of nearest neighbors each site has, and $\epsilon_{i-j}$ is the interaction between two sites occupied by entities of type $i$ and $j$ a distance $a$ apart where $i,j \in \{m, mem\}$. Membrane sites are notated with $mem$, while protein ones are indicated with $m$.

Each site-site interaction term can be determined using smaller scale MD simulations, or from elasticity models given the thickness profile [37,38]. However, in doing so, numerous unknown parameters and membrane characteristics are introduced to the system, such as lipid tilt or membrane tension. Note that since we neglect curvature its contribution to $\epsilon_m$ is not considered. Furthermore, the scale and nature of AFM data prevents the analysis of these quantities. We therefore treat $\epsilon_m$ as an effective parameter to be estimated.

We note that the energy difference between a discrete lattice with two proteins spread apart and one with them adjacent to each other is equivalent to Eq. 12. This energy difference can also be considered the energetic cost resulting from line tension and direct protein-protein interactions. These direct protein-protein interactions do not correspond to a defined structural interface, but rather a set of them, and is considered an effective oligomerization energy ($\epsilon_{olig}$). As a result, the effective interaction energy can be expressed as,

$$\epsilon_m = \epsilon_{olig} + 2a\gamma_m,$$ (13)

where $\gamma_m$ is the thinning-induced line tension and $\epsilon_{olig}$ is the energy in which these proteins are bound together. A more detailed derivation of this concept is shown in S2 Appendix.

## Finite difference method

To numerically solve the protein evolution equation shown in Eq. 6 and compare with the analytical cluster formation predictions, a finite difference method is utilized. The python code developed for this finite difference method can be found in [74]. Spatial derivatives are determined through fourth order central difference, while Fourth Order Runge Kutta is used for time evolution. Membrane (M) protein is initially distributed normally with an average equal to the initial protein density fraction ($\rho^*$) and a standard deviation of $5 \times 10^{-5}$, over a 250 x 250 grid. Each grid point is $0.4a$ in length, where $a$ is the approximate width of an M protein and is assumed to be $a = 5$ nm, while corresponding time steps depend on the effective interaction energy within the range of $[1 \times 10^{-9}, 5 \times 10^{-9}]$ s. High spatial and temporal resolution is required due to the lack of a more sophisticated finite element method. The temperature used throughout every simulation is $T = 303.15$ K, with the diffusion constant $A = 5 \times 10^{-13}$ m²/s.

To compare with the analytically determined wavevector and growth rate, a radial power spectrum of the density is calculated for every 100 time steps within linearity, defined as $\delta\rho < 0.01$. The maximum wavevector ($q_{max}$) is determined by fitting a Gaussian to the radial power spectrum, and equating it to the point in which there is a maximum. From here, the square root of this Gaussian at the maxima for each measured time step is fit to exponential growth,

$$\sqrt{|PS|} \sim e^{\omega_{0,max}t}.$$ (14)

Example power spectra and fits are shown in S3 Fig. Outside of linearity, the nearest neighbor distance is used to classify cluster formation periodicity. Clusters are defined according to a density threshold such that the protein area fraction is equivalent to the initial protein density ($\rho^*$). This process can be seen in S4 Fig for each of the images shown in Fig 4b. The evolution of nearest neighbor distance is shown in S5 Fig, and is measured every 10,000 time steps after $\rho_{max} > 0.9$. Time cut-offs for calculating nearest neighbor distance to minimize the impact of Ostwald ripening were found by determining the flattest region of the nearest neighbor distance evolution plots, with the cut-offs shown in S5 Fig.

## AFM sample preparation and imaging

The methods in this section are similar to those from [36]. First, monodisperse LUVs around 120 nm in diameter were prepared using a vesicle extruder, with lipid composition corresponding to that of the endoplasmic reticulum-Golgi intermediate compartment (ERGIC). The molar ratio of different lipids (Avanti Polar Lipids, Inc, Alabaster, AL) used are 1-palmitoyl-2-oleoyl-glycero-3-phosphocholine (POPC): 1-palmitoyl-2-oleoyl-sn-glycero-3-phosphoethanolamine (POPE): 1-palmitoyl-2-oleoyl-sn-glycero-3-phosphoinositol (POPI): 1-palmitoyl-2-oleoyl-sn-glycero-3-phospho-L-serine (POPS): Cholesterol = 0.45: 0.2: 0.13: 0.07: 0.15 [75]. The 5 mg/ml lipid solution in chloroform was dried in a glass vial with N2 gas stream, then vacuumed overnight at -30 in Hg. The dried lipid mixture was hydrated with buffer (150 mM NaCl, 20 mM HEPES, pH = 7.2) and 30 s vortex, prior to ten freeze-thaw cycles with dry ice and 37 °C bath. After the final thawing step, the aqueous solution was passed 11 times through a polycarbonate membrane with 100 nm pores (Nuclepore Track-Etch membrane, Whatman, Chicago, IL).

To reconstitute the M proteins into LUVs, concentrated stock solution (~ 400 mM) of n-dodecyl-$\beta$-D-maltoside (DDM Avanti Polar Lipids, Inc, Alabaster, AL) were added into 5 mg/mL of freshly extruded LUVs solution to reach a final concentration of 100 mM. M-protein stabilized by Triton X-100 (stock solution: 1 wt.% Triton X-100 per every 2 mg/mL M protein) was added to the LUVs solution at mass ratio of M/lipid = 1/100, 1/67, and 1/50 after 10 min of incubation. The solution was allowed another 10 min of incubation and the detergent was removed using 80 mg of wet BioBeads (Bio-Rad, Hercules, CA) per mL of LUVs. After 3 additions of BioBeads (once per 2 hours), the M-reconstituted LUVs were separated from the solution using the centrifuge.

For the preparation of supported bilayer samples for AFM imaging, 75 $\mu$L of the solution containing LUVs collected from the bottom of a microcentrifuge tube was deposited onto freshly cleaved pristine mica and incubated for 1 hour. Next, the sample was then gently rinsed with 5 mL of buffer. The samples were kept submerged by buffer during the sample preparation and imaging process. Imaging was performed within 1 hour of sample preparation.

An AFM fluid cell filled with buffer with an MSNL cantilever (Bruker, Camarillo, CA) in tapping mode was used. The spring constant was calibrated to be 0.30 N/m using the thermal oscillation spectrum. 512 x 512 pixel images with dimensions 2.25 $\mu$m x 2.25 $\mu$m were taken, with a vertical resolution of 0.01 nm and horizontal resolution of 4.4 nm. The cantilever tip size and image resolution was calibrated using 10 nm gold spheres (Ted Pella, Redding, CA) using our previously reported [36,76].

## Atomic force microscopy cluster determination

To define regions with and without protein from the AFM height images shown in Fig 2, a total variation denoising filter was used (TV Bregman from the python skimage library). The impact of this filter can be seen in S6 Fig. The TV Bregman filter allows for the maintenance of sharp height changes as the probe transitions from membrane to protein, while also reducing noise within protein clusters or large regions without protein [77,78]. After filtering the data for the case of Fig 2, regions with or without protein are decided based off a thresholding value defined using Otsu's method [79]. Anything above this value is a pixel with protein, while values below represent membrane. However, for the lowest area fraction, Otsu's method is not sufficient for protein determination, and thus the threshold value for the middle density fraction is used. Refer to S7 Fig for the dependence of threshold on area fraction.

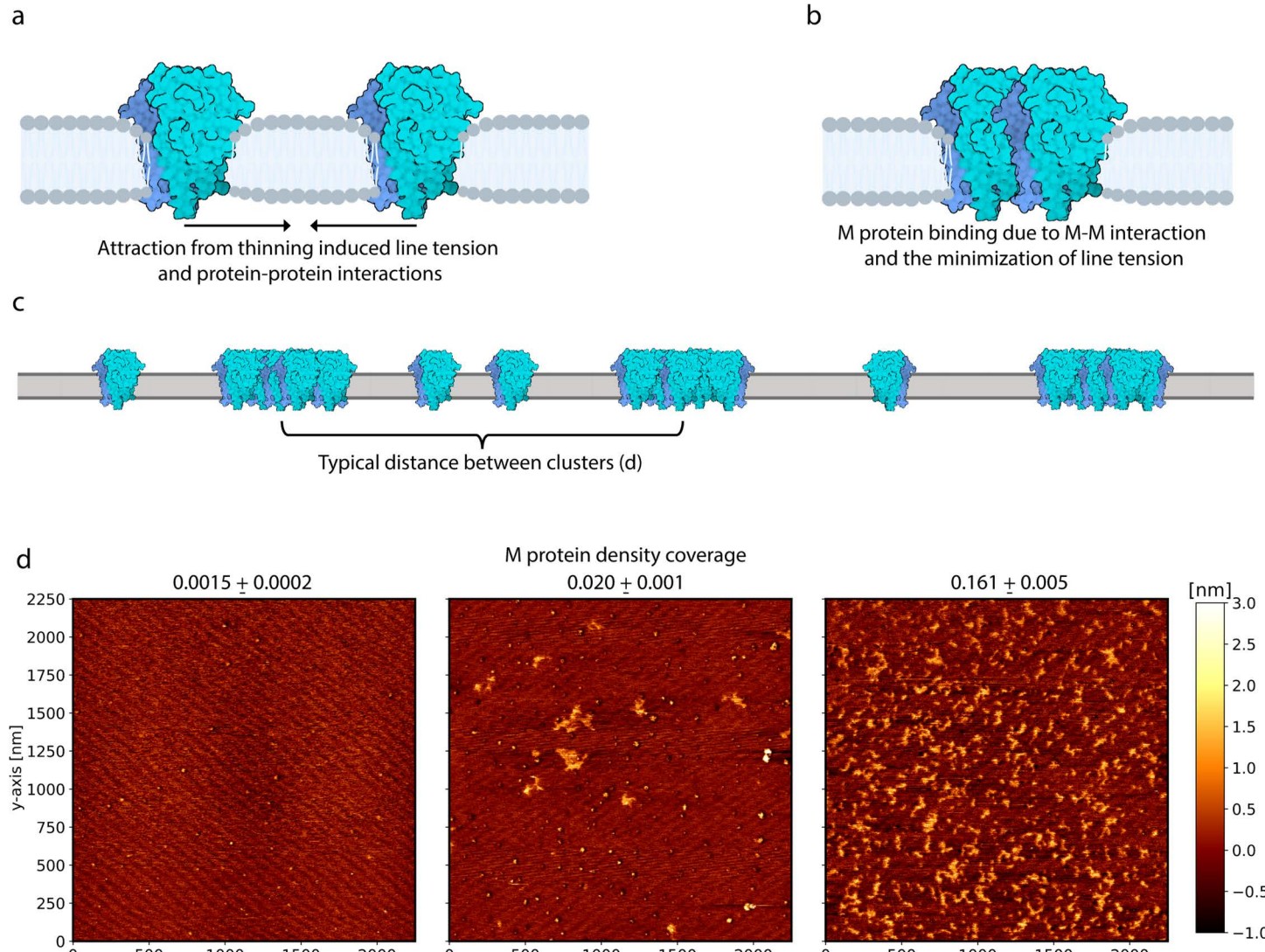

**Fig 2. Atomic force microscopy height images of 2.25 $\mu$m × 2.25 $\mu$m supported lipid bilayers showing cluster formation at different M protein area coverages. (a)** Cartoon showing nearby M protein attracting as a result of minimizing membrane-thinning-induced line tension. Longer range protein-protein interactions could also help this process. The different shades of blue represent the monomers within the dimer. **(b)** After attracting an adjacent protein, they will bind together as a result of direct M-M interactions and to further minimize line tension. **(c)** With sufficient protein density, enough of these complexes will form to create a periodic arrangement of clusters, defined by an average distance between them (*d*). Panels a-c were created with BioRender.com (https://biorender.com/ogth2uc). **(d)** AFM height images of a 2.25 $\mu$m × 2.25 $\mu$m supported membrane for three different M protein mass to lipid mass ratios. The corresponding protein density area coverage is shown for each ratio, increasing from left to right, where lighter regions represent greater heights.

Upon thresholding the filtered data, a cluster is defined as more than four adjacent pixels with protein. This restriction represents anything more than a single protein, since each pixel is approximately 4.4 nm across. After this restriction, the center of 'height' for each cluster is determined. This additional weighting accounts for proteins being off-center, or the pixel only showing the edge of the protein. Lastly, nearest neighbor distance is calculated such that each centroid is supplied a minimum distance representing the closest cluster using the sklearn python library NearestNeighbor function. For

two centroids that are both closest to each other, only a single value is accounted for. To determine whether the centroid density is roughly constant throughout the images, kernel density estimation from the sklearn python library is used (KernelDensity function). See S8 Fig for centroid density images at each protein area fraction.

## Results

### M protein induced thinning profile and line tension

To determine the short form's impact on membrane thickness we perform all-atom MD simulations of it embedded within a 25 nm x 25 nm lipid bilayer physiologically similar to the ERGIC. Final simulation frames, after 2 $\mu s$, can be seen in Fig 1a and Fig 1b, where the thickness is defined as the difference between the height of the upper and lower leaflets (Fig 1c). Averaged over the last 1.5 $\mu s$ of the simulation, the membrane thickness is shown from above in Fig 1d, with the cumulative cross-section of the protein in gold. While not completely axisymmetric, the membrane is noticeably thinner near the protein. Furthermore, taking a radial profile of membrane thickness by averaging over every angle with respect to the center of the protein leads to Fig 1e. Every bin is represented as a blue triangle, with the black trend line corresponding to averaging bins over a radial range. Additionally, the linear line of best fit and the corresponding 95% confidence interval in orange shows the significance of the short form's membrane thinning. Similar to the shorter MD results and AFM profile displayed in [36], the short form thins the membrane by 0.5 nm starting 12 nm from the center of the protein. Confirmation of the protein's stability can be seen in S9 Fig.

   We can compare the membrane thickness data to Eq. 1, which represents an analytic prediction of the membrane thickness as a function of distance from the center of the protein (purple line in Fig 1e). In our case, the thickness profile is parametrized by average monolayer thickness $\tau_0$ = 1.975 nm, unperturbed monolayer thickness $\tau_s$ = 2.1 nm, and total thinning $\delta$ = 0.25 nm. With the symmetric behavior of the membrane, each of these quantities are half of the full bilayer values. Integrating the elastic energy cost of this membrane thinning profile provides an estimate of the line tension, as described in Thinning induced line tension. Using elastic constants for a membrane from a similar system setup ($K_s \sim$ 3.0 $k_B T$/nm ± 1.0 $k_B T$/nm and $B_s \sim$ 3.0 $k_B T$ ± 1.0 $k_B T$) [36], the line tension from bending and tilt is approximately $\gamma_m$ = 0.10 $k_B T$/nm ± 0.04 $k_B T$/nm. We note that this line tension could serve as a possible membrane-mediated interaction to create clusters.

### M protein assembly depends on initial density fraction

Next, we use AFM to directly observe M protein cluster formation on a supported lipid membrane. First, shown in Fig 2a's cartoon, two nearby M proteins can attract solely in an attempt to minimize the elastic deformation of the membrane from thinning. As these proteins attract each other, Fig 2b displays a cartoon of the minimized line tension and role of direct M-M interactions in binding them together. Through these membrane-mediated interactions, protein clusters can form depending on the protein density with some characteristic distance between them (cartoon shown in Fig 2c).

   Using three different protein-lipid mass ratios provides three different protein area coverage values, shown as AFM images in Fig 2d for a 2.25 $\mu$m × 2.25 $\mu$m membrane with protein spread throughout. Area coverage is considered the percentage of pixels occupied with a protein, where protein occupation is defined based on a threshold AFM height value dependent on mass ratio. A mass ratio of 0.01 $\frac{M_M}{M_{lipid}}$ leads to the lowest area coverage, shown in the first panel of Fig 2d, ranging from 0.0013 to 0.0017 when accounting for the 0.01 nm vertical resolution of the AFM scan. In this case, M proteins are found individually or as small oligomers without any cluster formation. As this ratio is increased to 0.015 $\frac{M_M}{M_{lipid}}$ in Fig 2d panel 2, the area coverage increases to 0.020 ± 0.001. Furthermore, larger clusters begin to form while individual proteins or small oligomers remain isolated, leading to a non-isotropic protein density. When compared to the highest mass ratio of 0.02 $\frac{M_M}{M_{lipid}}$, the protein area coverage jumps to 0.161 ± 0.005, shown in Fig 2d panel 3. With this area coverage, protein clusters form roughly isotropically, where individual proteins are very rare, allowing for the characterization of

a typical distance between clusters ($d$). Thus, there exists a critical M protein density between the area coverages shown in Fig 2d panels 2 and 3, where clusters begin forming consistently with a characteristic distance. Position dependent centroid density for each of the three AFM images is shown in S8 Fig.

**Effective interaction energy defines onset of cluster formation**

To quantitatively understand the cluster formations in our planar AFM experiments and to better characterize M protein assembly overall, we utilize a Cahn-Hilliard model accounting for protein interactions through two-component regular solution theory described in Protein assembly continuum model. Linear stability analysis is used to identify the onset of cluster formation, providing parameter regimes in which these formations are possible (refer to Linear stability analysis).

Briefly, we derive a dispersion relation between the wavevector $q$ and corresponding growth rate $\omega_0$ of a perturbation. Unstable modes will have large positive values for $\omega_0$ and stable modes will have negative values. An example unstable relation is shown in Fig 3a, where there exists a maximum growth rate $\omega_{0,max}$ at wavevector $q_{max}$. This is considered the fastest growing mode and will dominate the others numerically and experimentally, leading to cluster formation with a spacing defined by the maximum wavevector, as seen in the upper right panel of Fig 3a. Additionally, a stable mode will decay back to the initial condition, or the proteins will be spread randomly (leftmost panel of Fig 2d and the bottom right panel of Fig 3a).

The effective interaction energy and initial protein density fraction define the onset of cluster formation. Initial M protein density fraction $\rho^*$ is equivalent to the AFM area coverage, ranging from 0 - 1. Fig 3b shows these parameters' impact on the maximum wavevector, where white represents a pairing without any cluster formation. As effective interaction energy is increased so is the maximum wavevector. Additionally, Fig 3c shows the maximum growth rate as a function of $\rho^*$ and $\epsilon_m$, which is asymmetric about $\rho^* = 0.5$ as opposed to the maximum wavevector.

Thus, for any $\epsilon_m$, there exists a range of $\rho^*$ where clusters will start to form. Furthermore, for any initial density fraction there exists a critical effective interaction energy at which the onset of assembly begins. This is in agreement with Fig 2d, where there exists some critical density at which consistent clusters begin to form.

Inverting the curves displayed in Fig 3b leads to a relation defining the effective interaction energy as a function of the initial protein density fraction and the maximum wavevector,

$$\epsilon_m = \frac{1}{\rho^* \left(1 - \rho^*\right) \left(1 - q_{max}^2\right)} \, k_\mathrm{B} T.$$

(15)

This can also be used to express the expected distance between clusters for a given interaction strength,

$$d = \frac{2\pi a}{\sqrt{1 - \frac{k_\mathrm{B} T}{\epsilon_m \rho^* \left(1 - \rho^*\right)}}}.$$

(16)

As a result, effective interaction energy can be estimated from our experimental AFM images shown in Fig 2d, assuming the relation holds beyond linearity. See S1 Appendix for the expanded version of maximum growth rate and maximum wavevector. Solely from the ability to form clusters at $\rho^* = 0.161$, a rough lower bound for the effective interaction energy can be identified: $\epsilon_m \sim 7.4 \, k_\mathrm{B} T$. However, direct comparison with AFM images using Eq. 15 is required for a more accurate estimation. While the onset of cluster formation is needed for further assembly, it is unknown whether these formations will survive beyond linearity. Thus, we numerically simulate the system within and outside of linearity to confirm Eq. 15 and determine whether these formations survive.

Numerical simulations of M protein density fraction evolution were performed with a finite difference method along a uniform grid (details provided in Finite difference method). Fig 4a shows the final frame within linearity for three different

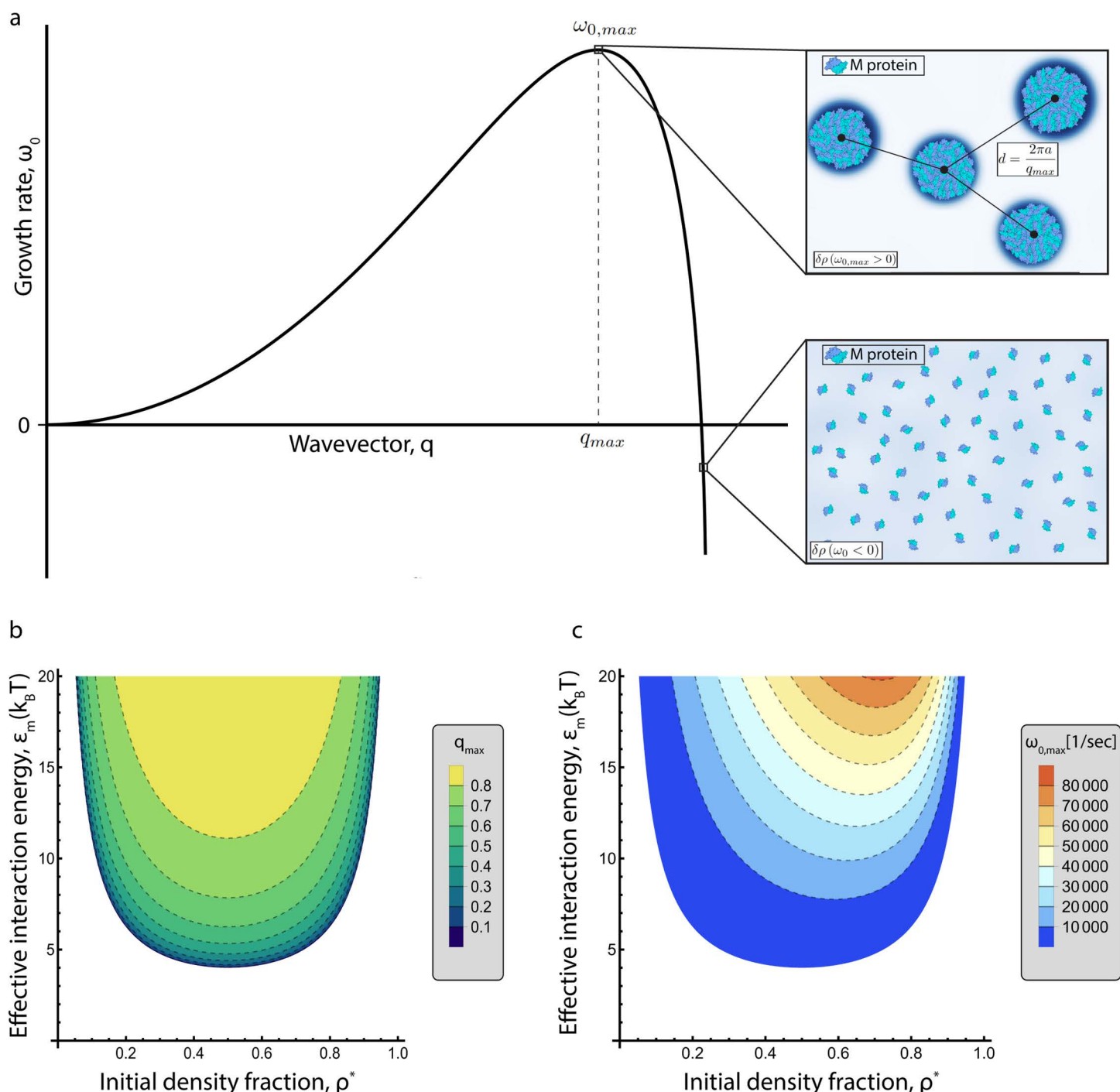

**Fig 3. Linear stability analysis of M protein assembly. (a)** Typical unstable dispersion relation, where each mode for the perturbation $\delta\rho$ is defined by its wavevector $q$ and its growth rate $\omega_0$. A growth rate greater than zero represents an unstable mode, where the fastest growing mode is described by $\omega_{0,max}$. Within this mode, M protein clustering will occur for an average distance between clusters $d$, as seen in the upper right panel. A stable mode is shown in the bottom right panel, where proteins remain isolated. Created with BioRender.com (https://biorender.com/geblf4d, https://biorender.com/90ou0jv). Phase diagrams for maximum wavevector **(b)** and maximum growth rate **(c)** with respect to the effective interaction energy ($\epsilon_m$) and the initial protein density fraction ($\rho^*$). Each dotted grey line represents a corresponding contour in the provided legend.

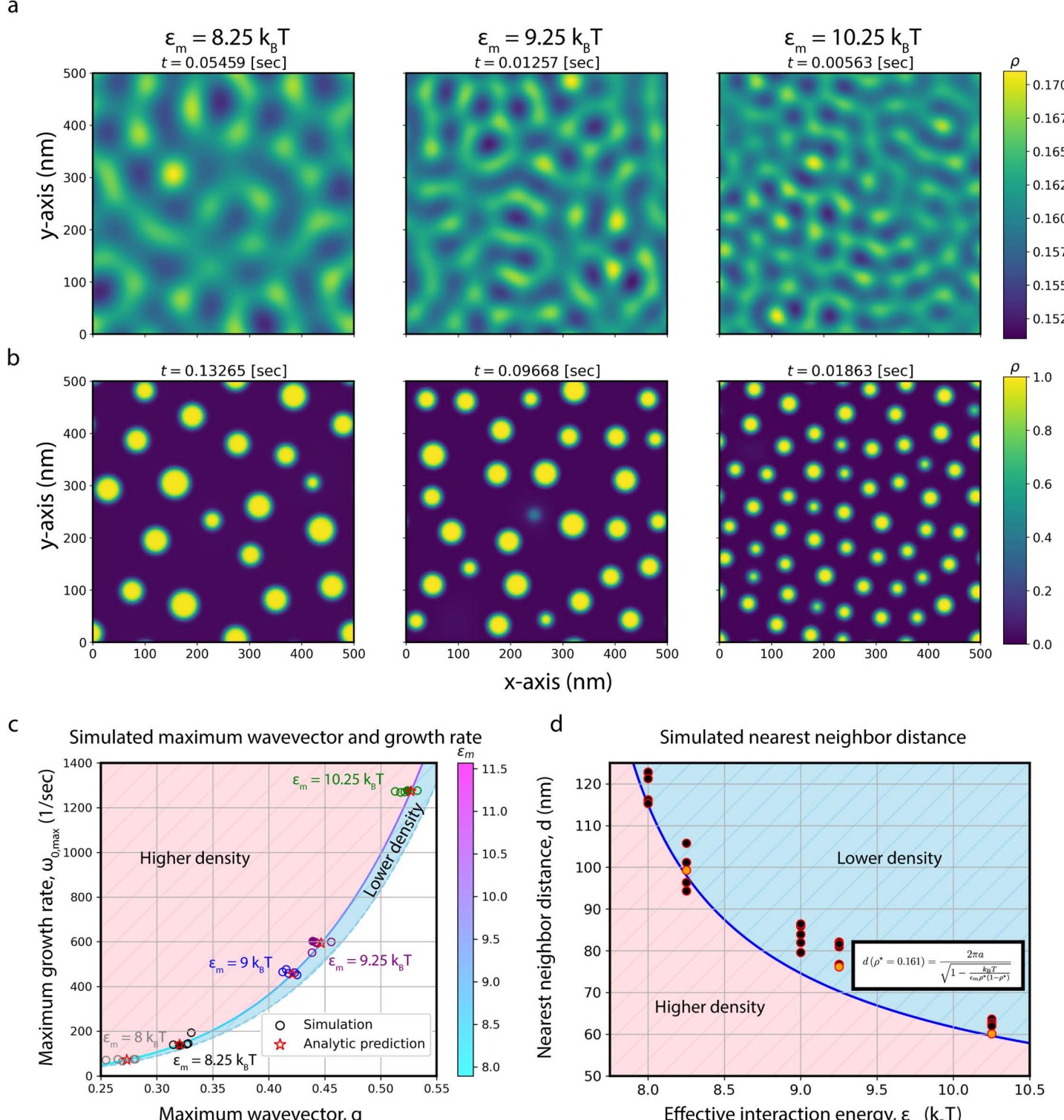

**Fig 4. Evolution of M protein density and clustering within and beyond linearity.** Simulations for three different interaction energies are shown for $\rho^* = 0.161$, with **(a)** portraying each at the final frame of linearity and **(b)** displaying later frames of a stable state before Ostwald ripening begins to dominate. The time of the frame is shown above the image, with brighter regions representing higher density, where the scale of the colorbar changes

between **(a)** and **(b)**. **(c)** Plot of the numerically determined maximum growth rate and maximum wavevector for the shown simulations and additional replicates within linearity. Each interaction energy is identified with a different color, where replicates are shown as hollow circles, those shown in **(a)** and **(b)** are filled in circles, and the analytic prediction for a given interaction energy is a red star. The analytic prediction for each interaction energy is shown with the line between the higher/lower density regions, which changes color according to $\epsilon_m$ in the corresponding colorbar. **(d)** Comparison between average nearest neighbor distance at stable states of numerical simulations to an analytic prediction for different interaction energies. Simulations shown in **(a)** and **(b)** are filled in orange, while extra replicates are filled with black. Representing the analytic prediction, the blue curve is determined using the highlighted equation. For **(c)** and **(d)**, the striped red region represents the direction the respective prediction would move if increasing density, while the striped blue region gives the reverse.

simulations at different effective interaction energies, with Fig 4b displaying a later time image prior to Ostwald ripening dominating the system for the same simulations. While density variations are smoothed out in Fig 4a, extrema appear roughly periodically, increasing in frequency as $\epsilon_m$ increases, agreeing with Fig 3b. Furthermore, the time needed to reach linearity decreases with increasing effective interaction energy. These trends are confirmed by calculating $q_{max}$ and $\omega_{0,max}$ for each simulation, as described in Finite difference method and shown in S3 Fig. Fig 4c shows these quantities within linearity, where individual simulations for each interaction energy are shown with an empty circle and simulations shown in Fig 4a are designated with a full circle. With the agreement between simulations and the analytic prediction (colored according to $\epsilon_m$, Eq. 25 in S1 Appendix), we confirm that our linear stability analysis holds in linearity.

## High density regions survive nonlinear transitions

The numerically determined evolution of M protein density from linearity to nonlinearity involves a sharp transition to a regime of diffusion limited growth where clusters gradually deplete the unclaimed surrounding protein. After depleting the reservoir of isolated protein, these clusters very slowly remodel, where they grow and shrink similarly to Ostwald ripening, typical of Model B dynamics [42]. Furthermore, through comparison between Fig 4a and 4b, it is clear that maxima within linearity survive cluster formation. This behavior can be seen for a variety of effective interaction energies in S1-S5 Videos, and for the shown systems in S10 Fig. Note that the small subset of clusters that form during the initial transition and quickly dissipate coincide to weaker maxima within the linear regime. With AFM formations seemingly within the diffusion limited growth regime (Fig 2d), analysis is only performed on simulations prior to Ostwald ripening dominating the system through a cut-off time. The cut-off time chosen for each simulation is shown above its x-y density plot, as seen in Fig 4b, with more information in Finite difference method and exact cut-off times shown in S5 Fig. Similar to linear stability analysis and Fig 4a, larger effective interaction energies decrease the time it takes for the system to reach the end of the diffusion limited growth dominated regime.

To quantify the survival of high density regions beyond linearity, the nearest neighbor distance between cluster formations within diffusion limited growth is used in combination with the maximum wavevector determined within linearity. Average nearest neighbor distance ($d$) is shown for each set of effective interaction energies in Fig 4d, where orange points correspond to simulations shown in Fig 4a and 4b and black to extra replicates. The analytic prediction, Eq. 16, from linear stability analysis is shown in blue, where the corresponding equation is highlighted. Spread between replicates for a given effective interaction energy are a result of the non-infinite size of the simulated box and the discrete number of clusters. Additionally, the slight overestimate of $d$, particularly for $\epsilon_m = 9/9.25\ k_B T$, stems from the initial stage of Ostwald ripening. Refer to S4 Fig for the process of calculating average nearest neighbor distance ($d$) with the images in Fig 4b. The evolution of this quantity, in addition to the number of clusters and density variance, for each effective interaction energy is shown in S5 Fig, where more information on nearest neighbor distance can be found in Finite difference method.

With the agreement in average cluster nearest neighbor distance for linear stability analysis and numerical simulations beyond linearity at a variety of effective interaction energies, Eq. 15 holds beyond linearity. Thus, it can be used to estimate effective interaction energy from AFM cluster formations. In Fig 4c and 4d, every $\epsilon_m$ has five replicates, where some points are so similar to the point of obscuring each other.

## Effective interaction energy from AFM images

Protein clusters are identified following the filtering and thresholding of the raw AFM data displayed in Fig 2d, as described in Atomic force microscopy cluster determination. Fig 5a shows the image with the highest area coverage, since clusters are not isotropically distributed otherwise, where red dots represent the centroid of a cluster. A representation of this cluster density for every area coverage is shown in S8 Fig. Binned nearest neighbor distances for each cluster are shown in Fig 5b, where the red curve represents a kernel density estimate of the histogram and the dotted red line signifies the corresponding maxima, equated to $d = 77.2$ nm. A comparable average nearest neighbor distance for a simulation with $\epsilon_m = 9$ $k_B T$ is shown through the black dotted line as $d = 79.6$ nm. This simulation is shown on the leftmost panel in Fig 5c and 5d.

With this nearest neighbor distance, the effective interaction energy can be calculated using Eq. 15. Accounting for the spread in the nearest neighbor distance 77.2 nm ± 1.2 nm, error in calculating area coverage 0.161 ± 0.005 (refer to S7 Fig), and the size of the protein $a \in [3.5, 5.5]$ nm, gives an effective interaction energy in the range of $\epsilon_m \in [7.8\ k_B T, 9.6\ k_B T]$. This relationship holds because AFM clustering appears to be within the diffusion limited growth regime.

Next, Fig 5c and 5d indicate the impact of higher initial protein density in cluster formation within our model. While linearity is still in agreement with our analytic predictions for $\rho^* = 0.161$, as seen in Fig 5c and S11 Fig, this is not the case for higher densities beyond linearity. For an initial density fraction of $2\rho^*$, shortly after transitioning beyond linearity, clusters begin to merge together. This leads to a much larger distance between clusters than the predicted quantity. In the case of $3\rho^*$, upon leaving linearity, lines of higher density are formed, with the typical cluster formations only appearing at much later time. In both cases, after sufficient time for stabilization, maxima within the linear regime do not coincide with clusters post transition. The transition beyond linearity for the middle and rightmost panels of Fig 5d involves coarsening and coalescence beyond diffusion limited growth. Clusters initially form and then merge together, most likely until there is only a single cluster left. These observations can be seen in S3, S6, and S7 Videos.

## Discussion

In this paper, we examined M protein cluster formation—an essential early step in the assembly and budding of SARS-CoV-2—across multiple length and time scales. As discussed in [36], the long-form M protein was not observed in our AFM experiments, consistent with the substantial structural rearrangements seen during long-form simulations [36] and the requirement for binding with a Fab complex for structural determination [28,35]. Whether the absence of the long form stems from the lack of additional viral structural proteins or its occurrence only within large, mature clusters remains unknown. Another possibility is that the conformational transition from the short to the long form involves a substantial energy barrier, making the long form inaccessible under the conditions of our simulations and AFM experiments. Thus, the cluster formation analyzed here is most representative of early stage assembly, involving only the short form, and does not include other aspects of viral assembly such as membrane curvature induction or interactions with additional structural proteins. Nucleation and energy barriers play an important role in viral assembly in general [80,81], and the emergence of short-form M clusters marks a key nucleation event that overcomes the initial energy barrier for assembly and sets the stage for subsequent curvature generation and budding. These clusters therefore represent not only the onset of viral assembly but also a crucial control point whose energetics could define bottlenecks in the budding process.

Utilizing all-atom MD of an individual short-form M protein embedded in an ERGIC-like membrane, we quantified the protein-induced membrane thinning on the scale of 0.5 nm near the protein, decaying over roughly 12 nm. The thinning appears purely elastic, without any noticeable correlation between location and lipid composition. As discussed in [36], this elastic deformation likely arises from the mismatch between the transmembrane domain (≈4 nm) and the average bilayer thickness. The thinning profile closely resembles that obtained in shorter MD runs of the same structure and AFM images of individual full-length M proteins [36], while also agreeing with previous observations of the SARS-CoV M protein [13]. Furthermore, the thinning-induced line tension derived from this deformation can be estimated as described in Thinning

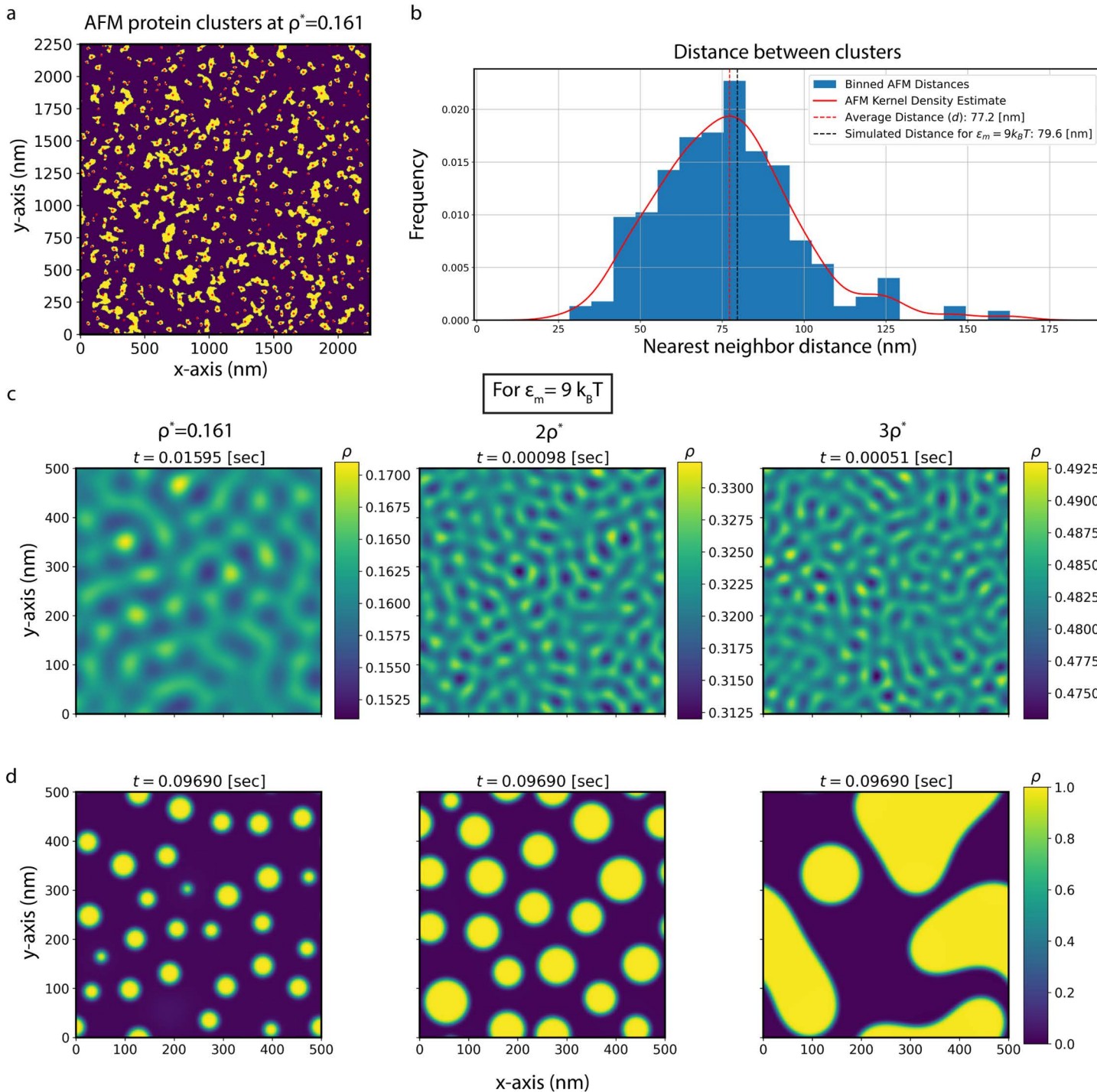

**Fig 5. Characterization of AFM cluster formation and comparison to numerical solutions of the continuum model at a variety of protein density fractions. (a)** Filtered and thresholded AFM image at the highest protein area coverage, $\rho^* = 0.161 \pm 0.005$. Protein clusters larger than four pixels are shown in yellow with their corresponding centroid in red, while other regions are in purple. **(b)** A normalized histogram of nearest neighbor distance between protein clusters is shown in blue, where the kernel density estimate is shown in red. The average nearest neighbor distance is taken as the point corresponding to the maximum in the kernel density estimate, and is shown as a red dotted line. A dotted black line displays a comparable $d$ for a simulation with $\epsilon_m = 9\ k_B T$. Simulation frames are shown at **(c)** the final frame of linearity and **(d)** $t = 0.0969$ s, with time at the top of each image. This time corresponds to the stable state before Ostwald ripening takes over for $\rho^* = 0.161$. Initial density fraction increases from left to right. Note that the colorbar in **(c)** changes for each figure. The simulated value shown in **(b)** is in leftmost column of **(c)** and **(d)**.

induced line tension. This line tension acts as a membrane-mediated attraction between neighboring M proteins with a magnitude of approximately 0.10 $k_\text{B}T$/nm ± 0.04 $k_\text{B}T$/nm.

To characterize M protein clustering, we analyzed AFM images of M proteins embedded in planar supported lipid bilayers. AFM images were acquired within a 2.25 $\mu$m × 2.25 $\mu$m membrane patch at three different protein area coverages (density fractions). Our results indicated a critical average density fraction above which isotropic clusters appeared, with formation observed at $\rho^* = 0.161 \pm 0.005$ but not at $\rho^* = 0.0015 \pm 0.0002$. While some clusters were observed for $\rho^* = 0.020 \pm 0.001$, their small sizes and heterogeneity suggest they likely result from local density variations or aggregation prior to vesicle insertion, rather than true phase-separated clusters at equilibrium. We attribute the lack of any observed Ostwald ripening at the highest imaged area fraction to the proteins being kinetically trapped, potentially due to interactions with the M protein's N-terminal and the bilayer supporting mica surface.

To quantitatively predict M protein cluster formation, we employed a continuum model describing density evolution on a flat membrane. Linear stability analysis revealed a critical density for cluster formation that depends on the effective interaction energy. Moreover, for each initial density fraction, a critical interaction energy exists above which clustering occurs. Based on the AFM data, we estimated a lower bound for the effective interaction energy of $\epsilon_m \sim 7.4$ $k_\text{B}T$. Linear stability analysis also allowed us to estimate this interaction energy directly from the average distance between clusters and the protein area coverage.

We then confirmed these predictions using finite-difference simulations within the density range observed in AFM data. In the linear regime, or for small density evolution, protein density appears as a set of smoothed extrema that gradually sharpen into high-density clusters located at the initial maxima for low-density fractions. This behavior is consistent with numerical solutions of similar models in the low-density regime [65]. Within this regime, linear stability analysis remains valid well beyond the linear phase, until late-time Ostwald ripening becomes dominant. For densities beyond that of our AFM data, where density fractions are two to three times higher, linear stability holds in the early regime but breaks down after clusters merge, leading to coalescence and coarsening not observed in AFM images.

Having established agreement between our linear stability analysis and numerical solutions beyond linearity, we compared theoretical predictions with AFM measurements to estimate effective interaction energies. This comparison only holds for low densities, where coalescence plays little to no role, and under the assumption that AFM measurements are kinetically trapped before Ostwald ripening dominates. From the AFM-determined average nearest-neighbor distance of 77.2 ± 1.2 nm between clusters, we estimated $\epsilon_m \in [7.8$ $k_\text{B}T$, 9.6 $k_\text{B}T]$. Using the MD-derived thinning-induced line tension of 0.10 $k_\text{B}T$/nm ± 0.04 $k_\text{B}T$/nm, we estimate thinning's contribution to the effective interaction energy as only [0.4 $k_\text{B}T$, 1.5 $k_\text{B}T$] dependent on the protein's variable width (see Effective interaction energy for details). Thus, membrane thinning alone does not play a dominant role in cluster formation, likely acting more prominently during later curvature generation and budding. The remaining portion of the effective energy, attributed to direct M–M interactions and considered an effective oligomerization energy, lies in the range $\epsilon_\text{olig} \in [6.9$ $k_\text{B}T$, 8.9 $k_\text{B}T]$. These energies are comparable to those of a model transmembrane protein observed with high-speed AFM, with membrane thinning introducing an attraction of $1-2$ $k_\text{B}T$ to a total interaction of ~6 $k_\text{B}T$ [39].

The higher-order oligomerization of M proteins has been consistently identified [13,28,36,41], providing support for the estimated effective oligomerization energy range. Adjacent dimers are also believed to predominantly contact each other through their C-terminal domains [13,28], as needed to achieve our detected tightly packed AFM clusters. While the structural interface for M protein higher-order oligomerization has not been elucidated, cryo-EM studies have extensively classified monomer-monomer contacts within the M protein dimerization interface for SARS-CoV-2 in its natural state [28,35], and when disrupted by small-molecules [40,41]. This interface remains comparable for other coronaviruses [82,83], with the C-terminal involved [28], and the transmembrane domain potentially dominating [84,85]. While our estimated effective oligomerization energy only considers interactions between dimers, the interface between monomers could play a role in the set of potential dimer-dimer contacts. Additionally, MD simulations of SARS-CoV-2 M protein structural predictions

previously identified a binding energy between two monomers similar in scale to our oligomerization energy [84], yet no studies have explored the dimer-dimer interface. Although it is unknown how these energies differ with M protein conformation, the act of altering conformations [25,26,40,41] is capable of inhibiting assembly, highlighting the importance of M–M interactions throughout the assembly and budding process [86].

Confirmation of linear stability predictions beyond the linear regime, together with the estimated interaction energies, yields a density fraction range of [0.118, 0.304] required for assembly-like cluster formation. To the best of our knowledge, M protein density along the ERGIC during budding has not been measured. However, by comparing the number of buds forming in a given compartment to the compartment's surface area, we can roughly estimate the minimal M protein density. Studies of coronavirus infected cells reveal the number of virions produced in an individual compartment at a given time varies widely [41,87–93], with smaller compartments ($r \sim 100$ nm) holding a single virion [89] and larger ones ($r \sim 300$ nm) capable of creating ten or more [87]. Given ~1000 M proteins with a radius of 2.5 nm in a single virion [13], and assuming a spherically shaped compartment, leads to an M protein area fraction lower bound of ~0.17 independent of compartment size, well within our predicted density range.

Although smaller apparent clusters are visible in AFM images at lower densities, these are unlikely to represent stable assembly intermediates. At higher densities, cluster coalescence may dominate, potentially disrupting the controlled assembly required for budding. However, given typical conditions inside cells, membranes are free to move and populated with numerous viral and host components. As such, curvature induction serves as a potential way to prevent this coalescence, while also introducing a membrane-mediated attraction between proteins capable of increasing $\epsilon_m$, and lowering the critical density and interaction range thresholds needed for assembly along a flat membrane [68,94,95]. Furthermore, additional viral components are also capable of lowering these critical thresholds, with E and M proteins known to colocalize [15,31,96]. Similarly, interactions between M and the RNA–N protein complex could prompt local M protein enrichment along the ERGIC surface, potentially lowering both thresholds [13,28,30,83]. While the overall clustering behavior remains robust for early budding stages, changes in membrane composition, tension, or temperature are also capable of modulating the critical density and interaction energy thresholds. Beyond shifting critical values, these behaviors play an even more important role at later budding stages.

Earlier models of SARS-CoV-2 assembly did not distinguish between the short and long forms of the M protein. Our results demonstrate that the short form alone can self-associate into clusters without other structural proteins. A plausible scenario is that assembly and budding begins with short-form cluster formation, followed by conformational transitions to the long form. These conformational transitions could be prompted by the oligomerization energy driving assembly, interactions with other structural proteins, or lipids [13,28,97]. Remaining short-form M proteins could localize near the bud neck, where their thinning effect facilitates membrane scission and virion release. The presence of such conformational and clustering-related energy barriers may also serve as kinetic control points, regulating the rate of assembly and budding. By modulating how quickly M proteins transition between conformations or coalesce into stable clusters, these barriers could fine-tune the timescale of virion formation, ensuring coordinated recruitment of other structural components.

Further work across multiple scales is needed to refine these conclusions. The estimated interaction energies—and particularly the contribution from direct M–M binding—can be validated using all-atom MD of multiple M proteins embedded in membranes, along with potential of mean force calculations. Such simulations would also allow for conformation-dependent interaction energies to be determined. Incorporating these results into continuum models that include curvature-dependent terms [64–68] would provide a more complete picture of how M proteins coordinate with other structural components to drive budding.

Our findings highlight that the short-form M protein alone can overcome the energetic bottleneck for initiating assembly and generate stable clusters through strong direct interactions. These insights identify quantitative thresholds—both in protein density and binding energy—that define the onset of viral assembly. From a therapeutic perspective, targeting these parameters could either provide new strategies for hindering SARS-CoV-2 replication, or explain previous ones

[40,41]. For example, reducing M–M binding energy through mutation or chemical interference below the critical threshold, or lowering the local M density below the required range, could effectively suppress assembly. Ultimately, a deeper understanding of M protein clustering and its energetic landscape provides a foundation for exploring novel antiviral strategies and offers broader implications for other enveloped viruses.

## Supporting information

**S1 Appendix. Nonlinear form of continuum model and conversion to linearity.**
(PDF)

**S2 Appendix. Approximating the contribution of line tension in effective interaction energy from a discrete protein lattice to a continuum.**
(PDF)

**S1 Fig. All-atom molecular dynamics atom counts (a) and simulation box size (b).** (a) Since the leaflets are symmetric, the number of molecules for a corresponding lipid type in a leaflet is half the system-wide value, adding to 1000 lipid molecules per leaflet. (b) After a quick change in the length along each axis within the first few nanoseconds, the box remains stable throughout the simulation.
(TIF)

**S2 Fig. Schematic of a discrete protein lattice with M proteins (a) far apart and (b) within range of nearest neighbor interactions.** The discrete protein lattice, with distance between lattice sites equivalent to the approximate width of the protein $a$, shows the two prominent types of site-site interactions when two proteins are not nearest neighbors: $\epsilon_{mem-mem}$ and $\epsilon_{m-mem}$. (b) As these proteins become nearest neighbors, $\epsilon_{m-m}$ encompasses direct interactions between proteins. Converting this representation for a system of randomly distributed proteins to a continuum results in the described continuum model. Created with BioRender.com (https://biorender.com/gxjhs1k, https://biorender.com/bkbr3e1).
(TIF)

**S3 Fig. Process for calculating maximum wavevector and growth rate from numerical simulations.** (a) Power spectra are shown for the three images displayed in Fig 4a, where the colorbar displays power spectrum density (PSD) dependent on wavelength per pixel. The radius of the prominent ring is the maximum wavevector. (b) After radially binning from the center of the spectra and averaging the PSD of non-unique radii, the radial profile as a function of the wavevector is shown for each effective interaction energy (wavevector is $\frac{2\pi a}{dl}$ times wavelength per pixel). Blue dots represent the average PSD for every distance from the center, while the black curve shows the corresponding gaussian fit with the maximum wavevector shown as the dotted orange line. (c) The square root of the PSD fit maxima for each measured time is shown with the dotted blue line, while the exponential growth fit is shown in red. The growth rate used for each of these plots is the corresponding maximum growth rate.
(TIF)

**S4 Fig. Thresholding and finding nearest neighbor distance for simulated cluster formation.** (a) Plot of the thresholded simulations at the cut-off time shown in Fig 4b, where regions with protein are shown in yellow and cluster centroids are shown in red. (b) Kernel density estimate for each of the three images in black ($\epsilon_m$ = 8.25 $k_BT$), purple ($\epsilon_m$ = 9.25 $k_BT$), and green ($\epsilon_m$ = 10.25 $k_BT$), where the dotted line displays the nearest neighbor distance in which there is a maximum ($d$). Each of these distances are shown as orange dots in Fig 4d.
(TIF)

**S5 Fig. Evolution of (a) simulated nearest neighbor distance, (b) number of clusters, (c) and variance for every simulation performed at $\rho^*$ = 0.161.** Each replicate is shown as a dotted line of variable color, where the corresponding

average of all replicates is shown as a black line for (a) and (b). Vertical cyan lines represent the cut-off time calculated by finding the time of minimal slope for the average nearest neighbor evolution curve. This time changes for each interaction energy, where $t_8 = 0.193$ s, $t_{8.25} = 0.13265$ s, $t_9 = .0969$ s, $t_{9.25} = .09668$ s, and $t_{10.25} = .01863$ s. The average value in (a) and (b) at the cut-off time is displayed with an orange dot. For (a), the horizontal cyan line represents the analytically predicted nearest neighbor distance. In (c), the slight fluctuations or bumps in density variance are a result of the gradually decreasing number of clusters stemming from Ostwald ripening shown in (b) for each interaction energy. None of these quantities were gathered until $\rho_{max} > 0.9$.
(TIF)

**S6 Fig. Using a total variation filter to reduce noise of raw AFM data while maintaining large protein clusters.** (a) The raw AFM data at the three different protein area coverages shown in Fig 2d. (b)-(c) This AFM data was filtered with total variation denoising using split-Bregman optimization, with a denoising weght of one and a tolerance of $1 \times 10^{-5}$. Height values for the cross-section shown in (a) and (b) as a dotted blue line are displayed in (c) for comparison between the raw and filtered data.
(TIF)

**S7 Fig. Thresholded and filtered AFM images for the three different protein area coverages.** Height thresholds for the two highest area coverages were found using Otsu's method, where the threshold can be found above each image in (a). Due to the low protein density in the leftmost image, its threshold was chosen to match that of the middle panel. (b) Protein area coverage is shown as a function of threshold height, where the used value is shown as a dotted vertical blue line. As the filter is applied, the curve gets closer to a step function, as expected for a transmembrane protein. The dotted grey line represents the higher bound on area coverage when considering AFM vertical resolution, while the dotted red line signifies the reverse, leading to the error bars in (a). Insets highlighting the change in area coverage near the chosen threshold heights are displayed for each system. Lastly, (c) displays a histogram of filtered image heights, with the threshold as a dotted black line. Note that Otsu's method loses effectiveness as the data becomes more Gaussian.
(TIF)

**S8 Fig. Cluster centroid density for each protein area coverage.** The top panel represents filtered and thresholded AFM data for each area coverage with cluster centroids shown as red dots. A cluster is defined as five or more adjacent pixels with height values greater than the threshold. Two-dimensional kernel density estimates of centroids are shown in the panel below, where higher density is shown in pink and lower is shown in blue. Of the three area coverages, only the highest has a close to constant density, showing isotropic cluster distribution.
(TIF)

**S9 Fig. The short form remains stable while embedded in an ERGIC-like membrane throughout the 2 $\mu s$ all-atom MD simulation.** (a) Root mean square deviation (RMSD) defines how much a protein structure has changed relative to its initial position. It is defined as $RMSD(t) = \sqrt{\frac{1}{N} \sum_{C_\alpha}^{N} \left[ r_i(t) - r_{i,0} \right]^2}$, where N is the total number of $C_\alpha$ atoms, $r_i(t)$ is the position for the i'th $C_\alpha$ atom at time $t$, and $r_{i,0}$ is the initial position of the corresponding atom. (b) Root mean square fluctuation (RMSF) is shown for both short form chains, where the N-terminal starts at residue 9 and the C-terminal ends at residue 204. RMSF is the averaged RMSD over time for each residue. (c) Total radius of gyration ($R_g$) of short form $C_\alpha$ atoms is shown in red. With little change throughout the simulation in (a) and (c), and reasonable RMSF for each residue in (b), the protein remains stable throughout the simulation. All of these quantities were calculated using the appropriate GROMACS command.
(TIF)

**S10 Fig. Later time Ostwald ripening shown through density evolution from (a) linearity to (b) end of diffusion limited growth dominated regime to (c) final simulation frame.** (a) Frames at the end of linearity from Fig 4a. (b) Frames

before Ostwald ripening dominates from Fig 4b. (c) Final simulation frames for corresponding interaction energies. Slight dissipation of clusters can be seen in each final frame, where smallest clusters in (b) or weaker maxima in (a) dissipate. Times are shown at the top of each image.
(TIF)

**S11 Fig. Maximum wavevectors and maximum growth rates during linearity agree with predictions for higher density simulations.** Each simulation was performed at $\epsilon_m$ = 9 $k_B T$, with five replicates for each of the three different initial protein densities: $\rho^*$ = 0.161 (blue), $2\rho^*$ (black), and $3\rho^*$ (green). Filled in circles correspond to replicates shown in Fig 5. Analytical curves from Eq. 25 in S1 Appendix, where the color gradient represents different effective interaction energies, are shown for each initial density fraction. Red stars are the expected analytic value for the maximum wavevector and maximum growth rate dependent on $\rho^*$. The lowest curve corresponds to $\rho^*$ = 0.161 at a variety of effective interaction energies, while the middle applies to $2\rho^*$, and the upper $3\rho^*$.
(TIF)

**S1 Video. Protein density evolution for $\rho^*$ = 0.161 and $\epsilon_m$ = 8 $k_B T$.**
(MP4)

**S2 Video. Protein density evolution for $\rho^*$ = 0.161 and $\epsilon_m$ = 8.25 $k_B T$.**
(MP4)

**S3 Video. Protein density evolution for $\rho^*$ = 0.161 and $\epsilon_m$ = 9 $k_B T$.**
(MP4)

**S4 Video. Protein density evolution for $\rho^*$ = 0.161 and $\epsilon_m$ = 9.25 $k_B T$.**
(MP4)

**S5 Video. Protein density evolution for $\rho^*$ = 0.161 and $\epsilon_m$ = 10.25 $k_B T$.**
(MP4)

**S6 Video. Protein density evolution for $\rho^*$ = 0.322 and $\epsilon_m$ = 9 $k_B T$.**
(MP4)

**S7 Video. Protein density evolution for $\rho^*$ = 0.483 and $\epsilon_m$ = 9 $k_B T$.**
(MP4)

## Author contributions

**Conceptualization:** Joseph McTiernan, Siyu Li, Umar Mohideen, Michael E. Colvin, Roya Zandi, Ajay Gopinathan.

**Data curation:** Joseph McTiernan, Yuanzhong Zhang.

**Formal analysis:** Joseph McTiernan, Yuanzhong Zhang.

**Funding acquisition:** Thomas E. Kuhlman, Umar Mohideen, Michael E. Colvin, Roya Zandi, Ajay Gopinathan.

**Investigation:** Joseph McTiernan, Yuanzhong Zhang, Ajay Gopinathan.

**Methodology:** Joseph McTiernan, Yuanzhong Zhang, Siyu Li, Thomas E. Kuhlman, Umar Mohideen, Michael E. Colvin, Roya Zandi.

**Project administration:** Umar Mohideen, Michael E. Colvin, Roya Zandi, Ajay Gopinathan.

**Resources:** Thomas E. Kuhlman, Umar Mohideen, Ajay Gopinathan.

**Supervision:** Umar Mohideen, Michael E. Colvin, Roya Zandi, Ajay Gopinathan.

**Validation:** Joseph McTiernan.

**Visualization:** Joseph McTiernan.

**Writing – original draft:** Joseph McTiernan, Yuanzhong Zhang, Umar Mohideen, Michael E. Colvin, Ajay Gopinathan.

**Writing – review & editing:** Joseph McTiernan, Yuanzhong Zhang, Siyu Li, Thomas E. Kuhlman, Umar Mohideen, Michael E. Colvin, Roya Zandi, Ajay Gopinathan.

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
