## [Decision Letter · Decision Letter 0]

11 Jan 2026

PCOMPBIOL-D-25-02364

Clustering of SARS-CoV-2 membrane proteins in lipid bilayer membranes

PLOS Computational Biology

Dear Dr. Gopinathan,

Thank you for submitting your manuscript to PLOS Computational Biology. After careful consideration, we feel that it has merit but does not fully meet PLOS Computational Biology's publication criteria as it currently stands. Therefore, we invite you to submit a revised version of the manuscript that addresses the points raised during the review process.

We look forward to receiving your revised manuscript.

Kind regards,

Arli Aditya Parikesit, PhD

Academic Editor

PLOS Computational Biology

Arne Elofsson

Section Editor

PLOS Computational Biology

**Additional Editor Comments:**

Based on reviewers' report, it is clear that extensive revision to the manuscript is imperative. Please kindly do the needful, and incorporate the letter of reply to the reviewers along with your revised manuscript.

**Journal Requirements:**

At this stage, the following Authors/Authors require contributions: Michael E. Colvin, Thomas E. Kuhlman, Siyu Li, Joseph McTiernan, Umar Mohideen, Roya Zandi, Yuanzhong Zhang, and Ajay Gopinathan. Please ensure that the full contributions of each author are acknowledged in the "Add/Edit/Remove Authors" section of our submission form.

3) We notice that your supplementary Figures are included in the manuscript file. Please remove them and upload them with the file type 'Supporting Information'. Please ensure that each Supporting Information file has a legend listed in the manuscript after the references list.

Potential Copyright Issues:

i) We note that Figure S2 is created through BioRender. Please confirm that you hold a Premium account and provide a pdf copy of the CC BY 4.0 Licence as provided by BioRender. For instructions on how to generate a CC BY 4.0 license for your figure, please see the guidelines here: https://help.biorender.com/hc/en-gb/articles/21282341238045-Publishing-in-open-access-resources.

If you are using the free assets from BioRender, we are unable to publish these images as they are licenced under a stricter licence than CC BY 4.0. In this case we ask you to remove the BioRender images and replace them with open source alternatives.

See these open source resources you may use to replace images / clip-art:

- https://bioart.niaid.nih.gov/

- https://bioicons.com/

- https://healthicons.org/

- https://scidraw.io/

- https://reactome.org/icon-lib

- https://www.phylopic.org/images

- https://journals.plos.org/plosbiology/article?id=10.1371/journal.pbio.3002395

6)  Please ensure that the funders and grant numbers match between the Financial Disclosure field and the Funding Information tab in your submission form. Note that the funders must be provided in the same order in both places as well.

**Reviewers' comments:**

Reviewer's Responses to Questions

**Comments to the Authors:**

Reviewer #1: This manuscript integrates long-timescale all-atom MD simulations, continuum modeling using a Cahn–Hilliard framework, and AFM imaging to quantify the interplay between direct M–M interactions and membrane-mediated forces in driving SARS-CoV-2 M protein clustering. The work aims to define the critical interaction energies and protein densities required for the onset of cluster formation, with implications for viral assembly mechanisms.

Overall, the study addresses a biologically important question and employs a rigorous multiscale approach. The manuscript is clearly written and technically sound, but certain aspects need clarifications, methodological justification, and better contextualization within coronavirus biology.

Major,

1, While the physics-based treatment is elegant, the biological implications are underdeveloped. The conclusion that “direct M–M interactions dominate over membrane-mediated interactions” is quantitatively supported, but the structural basis of such interactions is not discussed. Structural models of M dimers (short vs long forms) and known oligomerization interfaces should be incorporated into the interpretation.

2, The manuscript should discuss how the estimated interaction energies (≈ 7.8–9.6 kBT) compare to known coronavirus M protein oligomerization data, and viral assembly thresholds in other enveloped viruses.

3, The discussion would benefit from a clearer explanation of whether the derived critical densities are physiologically plausible in ERGIC membranes.

Minor,

1, Recent cryo-EM studies on coronavirus M oligomeric states should be more thoroughly cited.

Reviewer #2: This manuscript presents an important multiscale investigation of SARS-CoV-2 M-protein clustering using all-atom MD, continuum Cahn–Hilliard modeling, and AFM imaging. The topic is timely, the methods are generally robust, and the results offer quantitative insight into M–M interactions during early viral assembly. The work has clear potential for publication.

However, several issues require clarification before acceptance:

Continuum Model Assumptions:

The model neglects curvature–composition coupling despite the known curvature effects of M proteins (e.g., Introduction, lines 63–71). The authors should justify the flat-membrane approximation and discuss how it may shift critical interaction energies or densities.

Line Tension Estimation:

The thinning-induced line tension (~0.1 kBT/nm) relies on generic elasticity parameters rather than system-specific values. A sensitivity analysis or clearer limitations is needed.

AFM Area Coverage Determination:

Protein coverage estimates depend heavily on threshold selection (S7 Fig). The lowest-density image uses a threshold borrowed from another dataset, which may bias results. A robustness check is recommended.

Mismatch at Higher Densities:

Simulations at 2ρ* and 3ρ* show strong coarsening (Fig. 5c–d), unlike AFM images. The authors should discuss whether biological membranes exhibit mechanisms that arrest phase separation, which are not included in the model.

Applicability of Linear Stability Analysis:

The interaction energy ϵm derived from AFM spacing uses a linear-regime expression (Eq. 15), while AFM patterns are fully nonlinear. The authors should justify this extrapolation or clarify its limitations.

Despite these concerns, the manuscript is promising and would merit publication after major revision.

Reviewer #3: the review is uploaded as an attachment

**Have the authors made all data and (if applicable) computational code underlying the findings in their manuscript fully available?**

The PLOS Data policy requires authors to make all data and code underlying the findings described in their manuscript fully available without restriction, with rare exception (please refer to the Data Availability Statement in the manuscript PDF file). The data and code should be provided as part of the manuscript or its supporting information, or deposited to a public repository. For example, in addition to summary statistics, the data points behind means, medians and variance measures should be available. If there are restrictions on publicly sharing data or code —e.g. participant privacy or use of data from a third party—those must be specified.requires authors to make all data and code underlying the findings described in their manuscript fully available without restriction, with rare exception (please refer to the Data Availability Statement in the manuscript PDF file). The data and code should be provided as part of the manuscript or its supporting information, or deposited to a public repository. For example, in addition to summary statistics, the data points behind means, medians and variance measures should be available. If there are restrictions on publicly sharing data or code —e.g. participant privacy or use of data from a third party—those must be specified.

Reviewer #1: Yes

Reviewer #2: Yes

Reviewer #3: Yes

PLOS authors have the option to publish the peer review history of their article (what does this mean?). If published, this will include your full peer review and any attached files.). If published, this will include your full peer review and any attached files.

.

Reviewer #1: No

Reviewer #2: No

Reviewer #3: No

**Figure resubmission:**
---

## [Decision Letter · Decision Letter 1]

10 Apr 2026

Dear Prof. Gopinathan,

We are pleased to inform you that your manuscript 'Clustering of SARS-CoV-2 membrane proteins in lipid bilayer membranes' has been provisionally accepted for publication in PLOS Computational Biology.

Best regards,

Arne Elofsson

Section Editor

PLOS Computational Biology

Arne Elofsson

Section Editor

PLOS Computational Biology

Reviewer's Responses to Questions

**Comments to the Authors:**

Reviewer #1: The authors have adequately addressed all my concerns, and I have no further comments.

Reviewer #2: the manuscript can be accepted in this form

**Have the authors made all data and (if applicable) computational code underlying the findings in their manuscript fully available?**

The PLOS Data policy requires authors to make all data and code underlying the findings described in their manuscript fully available without restriction, with rare exception (please refer to the Data Availability Statement in the manuscript PDF file). The data and code should be provided as part of the manuscript or its supporting information, or deposited to a public repository. For example, in addition to summary statistics, the data points behind means, medians and variance measures should be available. If there are restrictions on publicly sharing data or code —e.g. participant privacy or use of data from a third party—those must be specified.requires authors to make all data and code underlying the findings described in their manuscript fully available without restriction, with rare exception (please refer to the Data Availability Statement in the manuscript PDF file). The data and code should be provided as part of the manuscript or its supporting information, or deposited to a public repository. For example, in addition to summary statistics, the data points behind means, medians and variance measures should be available. If there are restrictions on publicly sharing data or code —e.g. participant privacy or use of data from a third party—those must be specified.

Reviewer #1: Yes

Reviewer #2: Yes

PLOS authors have the option to publish the peer review history of their article (what does this mean?). If published, this will include your full peer review and any attached files.). If published, this will include your full peer review and any attached files.

.

Reviewer #1: No

Reviewer #2: No

---

## [Editor Report · Acceptance letter]

PCOMPBIOL-D-25-02364R1

Clustering of SARS-CoV-2 membrane proteins in lipid bilayer membranes

Dear Dr Gopinathan,

I am pleased to inform you that your manuscript has been formally accepted for publication in PLOS Computational Biology. Your manuscript is now with our production department and you will be notified of the publication date in due course.

With kind regards,

Anita Estes
